# Incidence and predictors of mortality among TB-HIV co-infected individuals on anti-tuberculosis and anti-retroviral dual therapy in Northwest Ethiopia: A retrospective cohort study

Abebe Fenta[1]*, Tebelay Dilnessa[1], Destaw Kebede[2], Mekuriaw Belayneh[1], Zigale Hibstu Teffera[1], Bewket Mesganaw[1], Adane Adugna[1], Wubetu Yihunie Belay[3], Habtamu Belew[1], Desalegn Abebaw[1], Bantayehu Addis Tegegne[3], Zelalem Dejazmach[4], Fassikaw Kebede[5], Gashaw Azanaw Amare[1]

**1** Department of Medical Laboratory Science, College of Medicine and Health Sciences, Debre Markos University, Debre Markos, Ethiopia, **2** Department of Medical Laboratory Science, Microbiology unit, Amhara Public Health Institute, Debre Markos, Ethiopia, **3** Department of Pharmacy, College of Medicine and Health Sciences, Debre Markos University, Debre Markos, Ethiopia, **4** Department of Medical Laboratory Science, College of Health Science, Woldia University, Woldia, Ethiopia, **5** Department of Public Health, College of Medicine and Health Sciences, Debre Markos University, Debre Markos, Ethiopia

\* abebefenta16@gmail.com, abebe_fenta@dmu.edu.et

## Abstract

### Background

Co-infection with the human immunodeficiency virus (HIV) and tuberculosis (TB) is a primary cause of death and morbidity. The rate of morbidity and death from TB-HIV is still Ethiopia's top health issue.

### Objective

This study aimed to assess the incidence and predictors of mortality among TB-HIV co-infected individuals on anti-TB and anti-retroviral dual Therapy at Debre Markos Comprehensive Specialized Hospital, Northwest Ethiopia.

### Methods

A retrospective cohort study was conducted at the Debre Markos Comprehensive Specialized Hospital among 436 TB-HIV co-infected individuals. A computer-generated random sampling technique was used to select patient charts registered from September 1st, 2011, and August 31st, 2020. Epi-Data version 3.1 was used for data entry, and STATA version 13 was used for the analysis. The Kaplan-Meier survival curve was applied to estimate the cumulative survival time of the TB-HIV patients. Log-rank tests were utilized to compare the survival time across various

**Data availability statement:** The data are available upon reasonable request on the corresponding author institution email addresses (dmuhsc@yahoo.com or HRCS@dmu.edu.et).

**Funding:** The author(s) received no specific funding for this work.

**Competing interests:** The authors have declared that no competing interests exist.

**Abbreviations:** AIDS, Acquired Immunodeficiency Syndrome; ART, Antiretroviral Therapy; BMI, Body Mass Index; CD4, Cluster of Differentiation 4; HIV, Human Immunodeficiency Virus; PLWH, People living with HIV; TB, Tuberculosis; WHO, World Health Organization

categories of explanatory variables. Bi-variable and multivariable Cox proportional hazard models were fitted to find predictors of TB-HIV mortality.

## Results

The mortality rate of TB-HIV co-infected individuals was 15.6%, with a median survival time of 42 months. Being male (Adjusted hazard Ratio (AHR)1.914;95%CI: 1.022–3.584), having CD4 count < 50 cells/mm3 (AHR 2.699; 95%CI: 1.220–5.973), being ambulatory (HR 2.794;95%CI: 1.459–5.352) and bedridden (AHR 5.514; 95%CI: 2.148–14.156), having low baseline weight (AHR 0.949;95%CI: 0.911–0.989), and having low hemoglobin level (AHR 0.927; 95%CI: 0.441–1.948) were important predictors for mortality.

## Conclusion and recommendation

The mortality rate among TB-HIV co-infected patients at Debre Markos Comprehensive Specialized Hospital was high. Being male gender, having a CD4 count below 50 cells/mm³, being ambulatory and bedridden, having low baseline weight, and having low hemoglobin were the important predictors of mortality. To reduce mortality, it is crucial to focus on the early identification and management of high-risk patients, particularly those with low CD4 counts, poor functional status, and low hemoglobin. Strengthening integrated TB and HIV care services is recommended to improve patient survival outcomes.

## Introduction

Tuberculosis (TB) and human immunodeficiency virus (HIV) have a bidirectional relationship, where each disease can accelerate the progression of the other in co-infected individuals. The two diseases worsen each other's effects, weakening the immune system. In areas with high rates of both diseases, HIV makes people more likely to develop active TB, either through new infections or reactivation of latent TB. Likewise, TB accelerates the progression of HIV to acquired immunodeficiency syndrome (AIDS) [1–3].

As stated by the World Health Organization (WHO) Global Tuberculosis Report 2023, HIV and TB are major public health challenges, contributing significantly to global morbidity and mortality [3,4]. The WHO 2023 global report recorded 7.5 million new TB cases, of which 0.63 million were co-infected with HIV. Despite standardized treatment for TB, it remains the leading cause of death from a single infectious agent, with 1.3 million deaths in 2022, including 0.17 million among people living with HIV (PLWH) [3,5]. The 2023 WHO report on tuberculosis in Africa states that 2.5 million people in the region were diagnosed with TB, and 500,000 died from it. Although Africa makes up only 15% of the global population, it accounts for 23% of new TB cases and 31% of deaths related to TB. The high HIV rate in the region is evident, with 20% of new TB cases occurring in PLWH and AIDS [6].

A global review found that TB-HIV co-infection caused 2.7 deaths per 100,000 people, with much higher rates in Lesotho (168 per 100,000) and Zambia (53 per 100,000) [7]. Overall, the mortality rate for people with both TB -HIV is 11%, with the highest rate in Africa at 14% [8]. In Sub-Saharan Africa, 36% of TB-HIV co-infection-related deaths are in children [9]. TB also accounts for 24.9% of deaths in hospitals among PLWH, placing a substantial financial burden on countries with high TB-HIV prevalence [10]. In a 2024 systematic review and meta-analysis by Fassikaw *et al.*, the overall TB-HIV mortality rate in Ethiopia was reported to be 16.2% [11].

To address the dual challenge of TB -HIV, the WHO has created a global framework for a strategic collaborative program and endorsed a test and treat all strategy [12]. The TB-HIV control programs are working harder to improve integrated services, case detection, TB prevention, and infection control, and provide both anti-TB and Antiretroviral Therapy (ART) medications [13,14]. Nonetheless, despite these initiatives and improved ART access in Ethiopia, the incidence of TB-HIV co-infection continues to rise, and the related illness and death rates have not decreased [14]. In general, HIV is the primary obstacle to achieving TB control targets in high-HIV settings, while TB remains a leading cause of death among PLWH [15]. In Ethiopia, managing individuals co-infected with TB-HIV is difficult due to the pill burden, heightened side effects, and potential drug interactions [16].

Previous studies have focused mainly on the survival rates of general HIV patients, giving less attention to TB-HIV co-infected individuals [17–19]. A few studies in Ethiopia have analysed predictors associated with mortality in HIV-infected TB patients. For instance, age [20–25], Sex [22,25,26], Poverty [27], Occupation [28], CD4, extra-tuberculosis [29–31], Anaemia [21], WHO stage [30–32] and Functional Status [30,32,33] were commonly reported predictors of mortality among TB-HIV co-infected individuals.

Countries, including Ethiopia, have adopted strategies to control TB-HIV co-infection, such as HIV counselling and testing, TB screening, and directly observed therapy for TB. These efforts aim to prevent both HIV and TB. However, despite early screening and diagnosis, TB-HIV co-infection remains a major health challenge in Ethiopia, especially in the study area. Therefore, this study aims to assess the incidence and factors predicting mortality among TB-HIV co-infected individuals receiving dual therapy at Debre Markos Comprehensive Specialized Hospital in Northwest Ethiopia.

## Methods and materials

### Study design, setting, and period

We conducted an institution-based retrospective cohort study at Debre Markos Comprehensive Specialized Hospital. The Hospital is in Debre Markos Town, in the Amhara Regional State, approximately 300 kilometres from the capital city, Addis Ababa, of Ethiopia. The hospital serves as the tertiary referral facility for a mostly rural agricultural area of more than 3.5 million people [34].The current study considered all TB-HIV co-infected clients between September 1, 2011, and August 31, 2020. The data were accessed from March 1 to April 30, 2022.

### Sample size estimation and procedure

Since the study was a cohort design, the required sample size for statistically significant results was calculated using a population proportion formula. The sample size was determined by considering key exposure variables and utilizing the Epinfo version 7 statistical package. Among these variables, WHO staging was selected as the primary exposure for non-accidental mortality because it provided the optimal sample size and yielded the most significant results. With a 5% significance level (two-sided), 80% power, and a 1:1 ratio of unexposed to exposed participants, the estimated mortality proportions in Ethiopia were 4.18% for the unexposed group (WHO stages I and II) and 12.5% for the exposed group (WHO stages III and IV) [30]. A total of 436 patient registration cards were randomly selected from the hospital's computerized register using a computer-generated simple random sampling method.

## Operational definitions

**Mortality.** Mortality refers to the death of a patient during the study period due to TB-HIV co-infection while receiving both Anti-TB and ART. Reports of death from clinicians, health agents, neighbours, or relatives are verified with healthcare providers and documented to ensure accuracy, excluding accidental deaths.

**Working.** The patient can perform job-related tasks and daily activities without significant limitations due to health.

**Ambulatory.** The patient can walk independently but may have limited ability to work or engage in some activities due to health issues.

**Bedridden.** The patient is confined to bed and requires assistance for most daily activities due to illness or severe health limitations.

**Eligibility criteria.** It is a medical guideline used to determine when patients should start treatment with ART. These criteria typically include the WHO clinical stage of HIV, CD4 count, or a combination of both.

**WHO clinical stage.** The WHO stage classifies HIV into four stages based on symptom severity and immune function. Stage I is asymptomatic or mild, stage II involves mild symptoms, stage III includes advanced symptoms, and stage IV is AIDS with severe infections or cancers.

**Body Mass Index (BMI).** It is a measure of body fat calculated by dividing weight (kg) by height (m²). It helps assess nutritional status, with categories ranging from underweight (BMI ≤ 18.5) to obesity (BMI ≥ 30), and is especially useful for individuals with conditions like TB and HIV.

**Adherence.** Adherence is considered good if it exceeds 95% (missing fewer than 2 doses out of 30 or fewer than 3 doses out of 60), as recorded by an ART physician; fair if it falls between 85% and 94% (missing 3–5 doses out of 30 or 3–9 doses out of 60), as recorded by an ART physician; and poor if it is below 85% (missing more than 6 doses out of 30 or more than 9 doses out of 60), as recorded by an ART physician.

## Study variables

**Dependent variable.** The dependent variable in this study was mortality, which was defined as the occurrence of death in individuals co-infected with TB and HIV.

**Independent variables.** The independent variables include socio-demographic factors such as sex (male, female), age category (15–30, 31–45, > 46–60), religion (Orthodox, Others), marital status (married, widowed, never married, divorced, separated), ethnicity (Amhara, Others), place of residence (rural, urban), education (no education, primary, secondary, tertiary), and occupation (merchant, non-governmental employed, governmental employed, day labourer, on job seeking, others). Clinical variables include WHO stage (I, II, III, IV), CD4 count (<50, 50–99, 100–200, 201–350, > 350), weight (20–29, 30–40, 41–50, 51–60, > 60), BMI (≤18.5, 18.5–25, ≥ 25), eligibility criteria (WHO clinical stage, immunological/CD4 count, both), drug regimen (AZT-based, D4T-based, TDF-based), hemoglobin (<10, ≥ 10), functional status (working, ambulatory, bedridden), TB type (pulmonary, extra-pulmonary), opportunistic infection (yes, no), substance use (yes, no), ART adherence (good, fair), and cotrimoxazole (given, not given).

## Data collection and quality control

A structured data extraction checklist was developed based on the national TB and ART registration guidelines, covering demographic, clinical, and treatment-related variables, and it was pre-tested on a 15% sample of patient records to ensure clarity and completeness.. The registers include the pre-ART register (registration of patients at their first visit), the ART register (registration after ART initiation), and the follow-up patient form. Both ART and TB registers have been looked through to record all persons who have been infected with HIV, started ART, and have been under an anti-TB treatment. One day of training was given to 5-degree holder nurse data collectors and the supervisor. A trained supervisor checked the completeness of the data to offer feedback to data collectors during the registration process and make corrections when needed.

 

## Data processing and analysis

Epi-data version 3.1 software was utilized for data entry, while Stata version 13 software was used for analysis. Descriptive statistics, including proportions, frequencies, and numerical summary measures of patient characteristics, were summarized. The Kaplan-Meier survival curve was utilized to estimate the survival time of TB-HIV co-infected individuals, and log-rank tests were used to compare the survival curves. A bi-variable Cox proportional hazards regression model was applied to each explanatory variable. Variables with a p-value ≤ 0.25 in the bi-variable analysis were then included in the multivariable Cox proportional hazards regression model. Hazard ratios with 95% confidence intervals and p-values were used to evaluate the strength of the associations and identify statistically significant predictors. In the multi-variable analysis, predictors with p-values < 0.05 were considered statistically significant for TB-HIV mortality.

## Ethical consideration

This study was conducted per the Declaration of Helsinki and received approval from the Research Ethics Committee of the College of Health Sciences, Debre Markos University, Ethiopia (CHS/ser/110/2022). Written informed consent was obtained from all participants. For minors, consent was obtained from their parents or guardians, and minors gave assent to participate. Illiterate participants had the consent form explained verbally to ensure understanding of the study's purpose, procedures, risks, and benefits. The participants then marked, or thumb printed in place of a signature, with a witness present to confirm the process. Ethical guidelines were followed when using data from deceased individuals, with consent obtained from their legal representatives, ensuring respect for the deceased and their families while maintaining privacy and confidentiality. A permission letter was also secured from the administrative authorities of Debre Markos Comprehensive Specialized Hospital. Patient information was kept confidential, and no personal identifiers were used on the data collection forms. Only the principal investigators had access to the data, which was kept secure and confidential.

## Results

### Socio-demographic characteristics

A total of 436 TB-HIV co-infected adult patient records were reviewed. Almost half of the reviewed records, 225(51.61%), were females, and nearly one-fourth, 107(24.5%), of them were single. Concerning the religious status majority, 401(92%) of them were orthodox Christian followers, and 202(46.3%) of the patients did not attend formal education. Almost all, 434(99.5%) of the total reviewed patient records were Amhara in ethnicity, 332(76.1%) were urban residents, and 107(24.5%) were non-government employed. The median age of the participants was 30 years (Table 1).

### Baseline clinical characteristics

Nearly three-fifths (73.8%) of the study participants had a baseline CD4 count ≤ 200 cells/mm$^3$. Additionally, 167(38.3%) patients were in the weight category of 41–50, and 137 (31.4%) were underweight (BMI ≤ 18.5). One-third (67.4%) of the participants were taking TDF ART regimens as a baseline drug. When ART and TB treatment were initiated, 21(4.8%), 72(16.5%), 239(54.8%), and 104(23.9%) patients had clinical WHO stages I, II, III, and IV, respectively. More than half, 257(58.9%) of the participants had pulmonary TB, and a few, 19 (4.4%) of the participants were bedridden on the functional status. Ninety-seven (22.2%) of the study participants were practicing substance abuse, while 120 (27.5%) of them were diagnosed with opportunistic infections (OI) during the baseline (Table 2).

### TB-HIV co-infection mortality rate

Within 42 (IQR: 0.5–89) months of the calculated median follow-up time, 15.6% (95% CI: 12.92, 18.83) of TB-HIV co-infected individuals died. Among the cohort, the mean age was 33.24 (standard deviation (SD) ±9.14), and the median weight at baseline was 50.00 kg (Inter Quartile Range (IQR) = 43.25 to 55.00 kg). The baseline median hemoglobin

**Table 1. Demographic factors of categorical covariates for TB-HIV co-infected individuals on anti-TB and antiretroviral dual therapy in western Amhara, Northwest Ethiopia.**

| Demographic Factors | | Frequency | Percentage |
|---|---|---|---|
| **Sex** | Male | 211 | 48.39 |
| | Female | 225 | 51.61 |
| **Age category** | 15-30 | 216 | 49.54 |
| | 31-45 | 175 | 40.14 |
| | >46-60 | 45 | 10.32 |
| **Religion** | Orthodox | 401 | 91.97 |
| | Others* | 35 | 8.03 |
| **Marital status** | Married | 186 | 42.7 |
| | Widowed | 37 | 8.5 |
| | Never married | 107 | 24.5 |
| | Divorced | 96 | 22. |
| | Separated | 10 | 2.3 |
| **Ethnicity** | Amhara | 232 | 99.5 |
| | Others | 2 | 0.4 |
| **Place of resident** | Rural | 104 | 23.9 |
| | Urban | 332 | 76.1 |
| **Education** | No education | 202 | 46.3 |
| | Primary | 114 | 26.1 |
| | Secondary | 93 | 21.3 |
| | Tertiary | 27 | 6.2 |
| **Occupation** | Merchant | 89 | 20.4 |
| | Non-Gov. Employed | 107 | 24.5 |
| | Governmental employed | 71 | 16.3 |
| | Day labourer | 101 | 23.2 |
| | On job seeking | 34 | 7.8 |
| | Others | 34 | 7.8 |

Others*: Muslim and Protestant.

was 12.60 (IQR = 11.00 to 13.60) and the baseline median CD4 count was 121.00 cells/mm$^3$ (IQR = 61.25 to 205.25) (Tables 3, 4). The Kaplan-Meier survival curve for the overall TB-HIV mortality rate is indicated in Fig 1. The curve shows a gradual decline in survival probability over time, indicating increased mortality as the duration of follow-up progresses.

### Log-rank test survival function of different predictor variables

In this study, males had a shorter survival time compared to females. The mean survival time for males was 6.4 months (SD ± 3.17). In comparison, for females it was 13.8 months (SD ± 2.31) (Fig 2). The mean survival time of TB-HIV co-infected patients having a working functional status was better than that of the other (bedridden and ambulatory). The survival time for working was 13.99 (SD + 4.32) while 5.34 (SD ± 1.2) for ambulatory and 4.54(SD ± 2.2) for bedridden (Fig 3).

### Bi-variable and multivariable Cox regression analysis

In the bi-variable Cox regression analysis being male, marital status, place of residence, TB type, functional status, WHO clinical stage, baseline CD4 count category, baseline weight category, AEC and drug regimen type, HGB category,

**Table 2. Baseline Clinical Characteristics for TB-HIV co-infected individuals on anti-TB and antiretroviral dual therapy in western Amhara, Northwest Ethiopia.**

| Baseline Clinical Characteristics | | Frequency | Percentage |
|---|---|---|---|
| WHO stage (N = 436) | I | 21 | 4.8 |
| | II | 72 | 16.5 |
| | III | 239 | 54.8 |
| | IV | 104 | 23.9 |
| CD4 count (N = 436) | <50 | 85 | 19.5 |
| | 50-99 | 93 | 21.3 |
| | 100-200 | 144 | 33 |
| | 201-350 | 83 | 19 |
| | >350 | 31 | 7.1 |
| Weight (N = 436) | 20-29 | 2 | 0.5 |
| | 30-40 | 81 | 18.6 |
| | 41-50 | 167 | 38.3 |
| | 51-60 | 148 | 33.9 |
| | >60 | 38 | 8.7 |
| BMI (N = 436) | ≤18.5 | 137 | 31.4 |
| | 18.5 −25 | 274 | 62.8 |
| | ≥25 | 25 | 5.7 |
| Eligibility criteria (N = 436) | WHO clinical stage | 97 | 22.2 |
| | Immunological/CD4 count | 87 | 20.0 |
| | Both | 252 | 57.8 |
| Drug regimen (N = 436) | AZT-based | 104 | 23.9 |
| | D4T-based | 38 | 8.7 |
| | TDF-based | 294 | 67.4 |
| Haemoglobin (N = 436) | <10 | 47 | 10.8 |
| | ≥10 | 389 | 89.2 |
| Functional status (N = 436) | Working | 256 | 58.7 |
| | Ambulatory | 161 | 36.9 |
| | Bedridden | 19 | 4.4 |
| TB type | Pulmonary | 257 | 58.9 |
| | Extra-pulmonary | 179 | 41.1 |
| Opportunistic infection | No | 316 | 72.5 |
| | Yes | 120 | 27.5 |
| Substance use | Yes | 97 | 22.2 |
| | No | 339 | 77.8 |
| ART adherence | Good | 433 | 99.3 |
| | Fair | 3 | 0.7 |
| Cotrimoxazole | Not given | 190 | 43.6 |
| | Given | 246 | 56.4 |

ART: antiretroviral therapy, BMI: Body Mass Index, CD4: Cluster of Differentiation 4.

substance use, OI infections, and adherence were identified as a candidate for multi-variable analysis. In the multivariable Cox regression analysis, only five variables were identified as predictors of mortality in individuals with TB-HIV co-infection. Accordingly, being male was 1.914 times (AHR: 1.914, 95%CI: 1.022, 3.584) more risk for death among TB-HIV

**Table 3. Descriptive statistics of the response variable for TB-HIV co-infected adult patients on anti-TB and antiretroviral dual therapy in Debre Markos Comprehensive Specialized Hospital, Northwest Ethiopia.**

| Patient Status | Continuous Variable | Mean | Standard deviation | Min. | Max. | Median | $Q_1$ | $Q_3$ |
|---|---|---|---|---|---|---|---|---|
| Death | Survival time | 16.04 | 18.83 | 1 | 83 | 7.5 | 2 | 24.75 |
| Censored | Survival time | 47.37 | 28.87 | 0.5 | 89 | 53.5 | 18.25 | 75 |
| Overall | Survival time | 42.48 | 29.78 | 0.5 | 89 | 42 | 13 | 71.75 |

**Table 4. Descriptive statistics of continuous variables for TB-HIV co-infected individuals on anti-TB and antiretroviral dual therapy in western Amhara, Northwest Ethiopia.**

| Characteristics | | Age | Baseline CD4 count | Baseline weight | baseline BMI | Baseline Hgb |
|---|---|---|---|---|---|---|
| **Mean** | | 33.24 | 155.56 | 49.83 | 20.55 | 12.35 |
| **Median** | | 31.00 | 121.00 | 50.00 | 20.00 | 12.60 |
| **Mode** | | 30 | 87.00 | 40.00 | 18.50 | 12.600 |
| **Std. Deviation** | | 9.145 | 131.03 | 9.30 | 2.974 | 1.86 |
| **Percentiles** | 25 | 27.00 | 61.250 | 43.25 | 18.50 | 11.00 |
| | 50 | 31.00 | 121.00 | 50.00 | 20.00 | 12.60 |
| | 75 | 38.00 | 205.25 | 55.00 | 22.50 | 13.60 |

BMI: Body Mass Index, Hgb: hemoglobin.

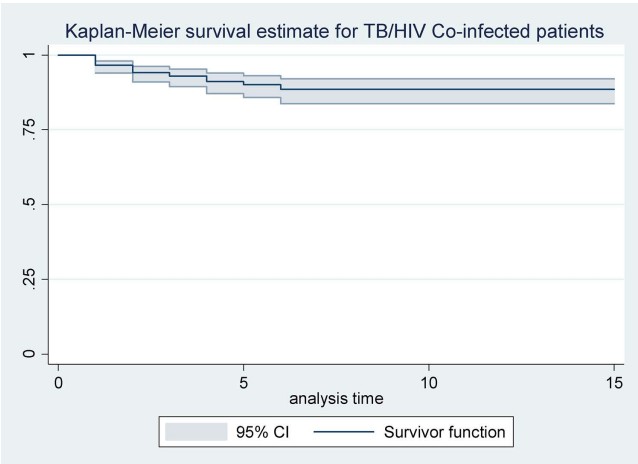

**Fig 1. The plot of the overall estimate of Kaplan-Meier survivor function of TB-HIV co-infected individuals on anti-TB and antiretroviral dual therapy in Debre Markos Comprehensive Specialized Hospital, Northwest Ethiopia.**

co-infected individuals compared with females. Similarly, having a baseline CD4 count <50 cells/m³ was 2.699 times (AHR: 2.699, 95%CI: 1.220, 5.973) more risk of dying from TB-HIV co-infected compared with individuals having a CD4 count of greater than 350 cells/m³. In addition, being ambulatory in functional status was 2.794 times (AHR: 2.794, 95%CI: 1.459, 5.352) more at risk of dying from a TB-HIV co-infected individual compared with the counterpart (working). Moreover, being bedridden in functional status was 5.514 times (AHR: 5.514, 95%CI: 2.148, 14.156) more risk of dying from a TB-HIV co-infected individual compared with the counterpart (working). In this study, we found that a 1 kg increase in

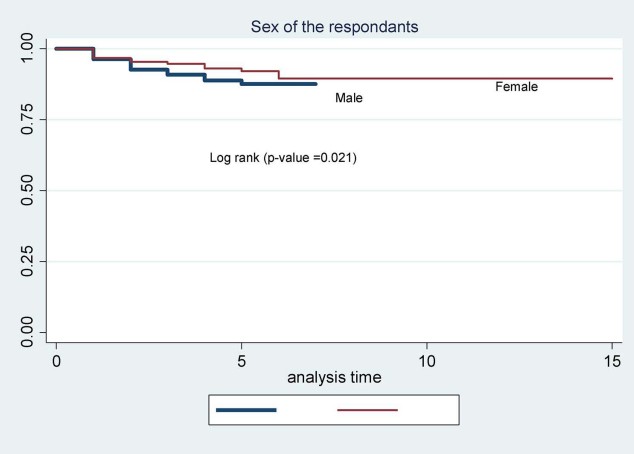

**Fig 2. Kaplan-Meier survival curves to compare the survival time of TB-HIV co-infected individuals on anti-TB and antiretroviral dual therapy, adults with different categories of sex in Debre Markos Comprehensive Specialized Hospitals, Northwest Ethiopia.**

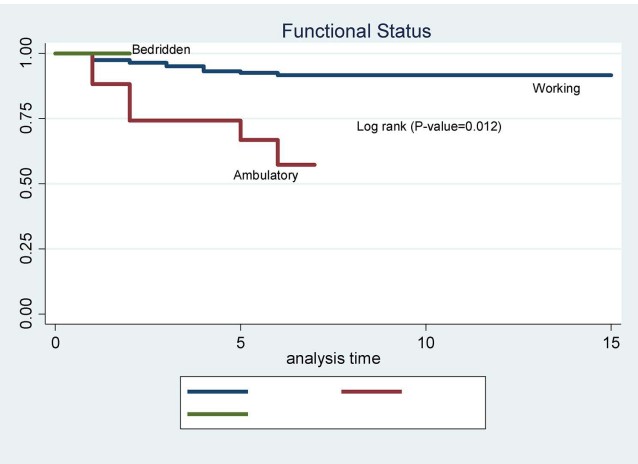

**Fig 3. Kaplan-Meier survival curves to compare the survival time of TB-HIV co-infected individuals on anti-TB and antiretroviral dual therapy, adults with different categories of functional status in Debre Markos Comprehensive Specialized Hospitals, Northwest Ethiopia.**

baseline weight of TB-HIV co-infected individuals reduces the risk of death by 0.949 (AHR: 0.949, 95% CI: 0.911, 0.989). Similarly, a 1 kg increase in hemoglobin levels was associated with a 0.927 reduction in the risk of death (AHR: 0.927, 95% CI: 0.441, 1.948) (Table 5).

## Discussion

Despite the widespread implementation of anti-TB and antiretroviral therapies, TB and HIV still rank as the leading causes of death among co-infected individuals, especially in low- and middle-income countries [35]. Due to its high mortality and morbidity rates, TB-HIV co-infection is considered a public health concern in Ethiopia [22,24,36]. This institution-based, retrospective cohort study aimed to assess the incidence and factors predicting mortality among TB-HIV co-infected individuals on dual therapy in the Western Amhara region. The study found a 15.6% mortality rate

**Table 5. Estimated values of the coefficients adjusted hazard ratios, 95% CI for the adjusted hazard ratio, and P-values of the final Cox PH model for TB-HIV co-infected individuals on anti-TB and antiretroviral dual therapy in Debre Markos Comprehensive Specialized Hospital, Northwest, Ethiopia.**

| Variable | | Survival status | | CHR (95% CI) | P value | AHR (95% CI) |
|---|---|---|---|---|---|---|
| | | Died | Censored | | | |
| **Sex** | Male | 42 | 169 | 1.456 (0.71, 2.985) | 0.042* | 1.914(1.022, 3.584) |
| | Female | 26 | 199 | 1 | | 1 |
| **RESIDENT** | Rural | 21 | 83 | 1 | | 1 |
| | Urban | 47 | 285 | 1.171(0.521, 2.631) | 0.299 | 1.366(0.758, 2.461) |
| **WHO STAGE** | I | 0 | 21 | 1 | | 1 |
| | II | 4 | 68 | 0.01(0.001, 7.251) | 0.521 | 0.421(0.030, 5.893) |
| | III | 28 | 211 | 0.214(0.051, 0.899) | 0.543 | 0.082(0.003, 4.27) |
| | IV | 36 | 68 | 3.520(2.137,10.285) | 0.170 | 2.197(1.236, 3.907) |
| **CD4 category** | <50 | 25 | 60 | 5.594(1.484, 21.089) | 0.014* | 2.699(1.220, 5.973) |
| | 50-99 | 15 | 78 | 10.681(2.543, 44.871) | 0.106 | 8.681(1.958, 38.488) |
| | 100-200 | 10 | 134 | 4.915(1.306, 18.53) | 0.281 | 1.599(0.681, 3.757) |
| | 201-350 | 12 | 71 | 3.171(0.793, 12.684) | 0.372 | 4.394(1.615, 11.952) |
| | >350 | 6 | 25 | 1 | | 1 |
| **Baseline wt.** | 20-29 | 2 | 0 | 1 | | 1 |
| | 30-40 | 14 | 67 | 1.807(0.506, 5.106) | 0.465 | 0.768(0.378, 1.560) |
| | 41-50 | 27 | 140 | 0.634(0.507, 0.794) | 0.551 | 0.826(0.440,1.550) |
| | 51-60 | 22 | 126 | 0.94(0.908, 0.973) | 0.028* | 0.949 (0.911, 0.989) |
| | >60 | 3 | 35 | 0.483(0.306, 0.693) | 0.678 | 0.762(0.211, 2.751) |
| **HGB category** | <10 | 13 | 34 | 1 | | 1 |
| | =>10 | 55 | 334 | 1.189 (0.361, 3.921) | 0.040* | 0.927(0.441, 1.948) |
| **OI Category** | Yes | 32 | 284 | 0.49647, (0.863, 2.656) | 0.228 | 1.485(0.781, 2.826) |
| | No | 36 | 84 | 1 | | 1 |
| **EC** | Clinical | 10 | 87 | 0.828(0.346(0.364,1.877) | 0.120 | 0.167(0.038, 0.743) |
| | Immunological | 4 | 83 | 0.232(0.054, 0.992) | 0.993 | 3.721(0.0101,41.341) |
| | Both | 54 | 198 | 1 | | 1 |
| **FEATURES** | Working | 17 | 239 | 1 | | 1 |
| | Ambulatory | 40 | 121 | 4.338 (1.947, 9.664) | 0.002* | 2.794(1.459, 5.352) |
| | Bedridden | 11 | 8 | 7.765 (2.101, 28.694) | 0.001* | 5.514(2.148, 14.156 |
| **SU** | Yes | 25 | 72 | 0.553(0.220(0.253,1.207) | 0.326 | 0.698(0.341, 1.43) |
| | No | 43 | 296 | 1 | | 1 |

CHR: Crude hazard ratio, AHR: Adjusted hazard ratio, HGB: Haemoglobin, OI: opportunistic infection, wt.: weight, EC: eligibility criteria, Fstatus: Functional status, SU: Substance use, *: statistically significant at p-value <0.05.

(95% CI: 12.92, 18.83) among TB-HIV co-infected adults at Debre Markos Comprehensive Specialized Hospital. This result is in agreement with studies from Northwest Ethiopia (18%) [33], Sub-Saharan Africa (18.1%) [37], and Myanmar (13.7%) [38].

Moreover, the present study found a lower TB-HIV mortality rate compared with a study finding reported in Southwest Ethiopia (20.2%) [23], China (34.7%) [39], and Brazil (27.4%) [40]. This difference in TB-HIV mortality rates may be due to variations in sample size, study period, and socio-demographic factors. Ethiopia's lower mortality, despite being a developing country, is likely due to targeted health programs, community support, early treatment, and international aid. In contrast, higher HIV rates and more comorbidity in China and Brazil contribute to higher mortality.

In another way, our findings revealed a fairly higher mortality rate among TB-HIV co-infection individuals when compared to a study carried out in Addis Ababa, Ethiopia, India, and Thailand [32,41,42]. The observed difference between the present and previous studies may be due to the variation in socio-demographic factors and the residence of the study population.

Similarly, the current study also identified predictors of TB-HIV co-infected adult mortality rate and found sex, CD4 count, functional status, baseline weight, hemoglobin, and eligibility criteria. Being male was more (AHR: 1.914, 95%CI: 1.022, 3.584) risk of dying due to TB-HIV co-infections compared with the counterpart. In contrast, this result was not supported by a study result reported in southern Ethiopia, where females are more at risk for a high rate of TB-HIV mortality [43]. The higher risk of TB-HIV mortality in males could be due to weaker immune responses. Viral infections are generally more common and severe in males, while females tend to have stronger immune responses, which lead to faster virus clearance but also a higher risk of immune-related damage [44,45]. Additionally, behaviors like smoking and alcoholism, which are more common in males, may contribute to the higher risk of TB-HIV mortality [46,47].

This study also found that TB-HIV co-infected individuals with a CD4 count less than 50 have a significantly higher risk (AHR: 2.699, 95% CI: 1.220, 5.973) of adverse outcomes compared to those with a CD4 count greater than 350. Previous studies conducted in Northwest Ethiopia, Burkina Faso, Zimbabwe, and Ukraine support these findings, indicating that a lower CD4 count increases the risk of death in TB-HIV co-infected individuals [33,48–50].

Moreover, in this study, individuals with an ambulatory functional status had a threefold higher likelihood of death compared to those who were working (AHR: 2.794, 95%CI: 1.459, 5.352). Additionally, bedridden TB-HIV clients faced a significantly higher risk of death (AHR: 5.514, 95%CI: 2.148, 14.156) compared to those who were working. This finding is supported by previous studies conducted in Northwest Ethiopia and Ambo, Ethiopia, which showed that being bedridden and ambulatory increases the risk of death compared to being working functional status [30,33]. Furthermore, our findings align with a previous study that indicated bedridden functional status is associated with a higher risk of death for TB-HIV co-infected individuals compared to those who are working [43].

In this study, we found that for every 1 kg increase in baseline weight, the risk of death from TB-HIV co-infection decreased by 0.949 (AHR: 0.949, 95%CI: 0.911, 0.989). This finding is consistent with previous studies conducted in Ambo, Ethiopia [30], Northwest Ethiopia [25], and other evidence [20–24,33]. This might be due to the increased weight is often associated with a better quality of life, improved treatment outcomes, and higher daily intake of protein and energy. People with TB-HIV typically experience a loss of appetite, which leads to weight loss [51]. Additionally, individuals who gain weight may consume more protein and energy, which can strengthen the immune system and reduce TB-HIV-related mortality [52].

On the other hand, this study found that a 1 kg increase in hemoglobin levels for TB-HIV co-infected individuals reduces the risk of death by 0.927(AHR: 0.927, 95%CI: 0.441, 1.948). This may be because low hemoglobin levels are an indication of disease complications and a higher risk of death. Evidence shows that the severity of the disease can lower hemoglobin levels, and this decrease in hemoglobin is associated with an increased risk of TB-HIV mortality [52,53].

## Conclusions and recommendations

The mortality rate among TB-HIV co-infected individuals at Debre Markos Comprehensive Specialized Hospital was notably higher than in previous studies conducted at the same facility. Key predictors of TB-HIV mortality included being male, having a CD4 count below 50, poor functional status (such as being ambulatory or bedridden), low baseline weight, and low hemoglobin levels. To reduce mortality, it is important to regularly monitor CD4 counts and hemoglobin levels, improve nutrition and weight management, and provide special attention to male patients and those with low functional status. Additionally, strengthening early detection, treatment adherence, and management of TB-HIV co-infection is essential to lowering mortality rates.

## Strengths and limitations of the research

The study benefits from a large 10-year dataset, standardized national registers, and a long follow-up period, enabling robust assessment of mortality predictors. Limitations include potential missing or incomplete data, limited information on some key predictors, possible register errors, and being a single-center study, which may affect generalizability.

## Supporting information

**S1 File. Data extraction checklist (supplmentary file 1).**
(DOCX)

## Acknowledgments

We would like to forward our gratitude to Debre Markos University, College of Health Science, for providing us with the ethical letter. Then we would like to acknowledge also Debre Markos Comprehensive specialized Hospital administrative bodies for permitting the patient records. Finally, we would like to express our sincere gratitude to the data collectors for their dedicated work and for submitting the required data promptly.

## Author contributions

**Conceptualization:** Abebe Fenta, Destaw Kebede, Bantayehu Addis Tegegne, Zelalem Dejazmach.

**Data curation:** Abebe Fenta, Zigale Hibstu Teffera, Bewket Mesganaw, Desalegn Abebaw, Zelalem Dejazmach.

**Formal analysis:** Abebe Fenta, Zigale Hibstu Teffera, Adane Adugna, Habtamu Belew, Bantayehu Addis Tegegne, Fassikaw Kebede.

**Funding acquisition:** Abebe Fenta, Adane Adugna, Wubetu Yihunie Belay, Desalegn Abebaw, Bantayehu Addis Tegegne.

**Investigation:** Abebe Fenta, Destaw Kebede, Mekuriaw Belayneh, Adane Adugna, Habtamu Belew, Desalegn Abebaw, Bantayehu Addis Tegegne, Zelalem Dejazmach, Fassikaw Kebede.

**Methodology:** Abebe Fenta, Tebelay Dilnessa, Destaw Kebede, Mekuriaw Belayneh, Zigale Hibstu Teffera, Wubetu Yihunie Belay, Habtamu Belew, Bantayehu Addis Tegegne, Zelalem Dejazmach, Fassikaw Kebede.

**Project administration:** Abebe Fenta, Tebelay Dilnessa, Gashaw Azanaw Amare.

**Resources:** Zigale Hibstu Teffera, Bewket Mesganaw, Adane Adugna, Desalegn Abebaw, Bantayehu Addis Tegegne.

**Software:** Abebe Fenta, Tebelay Dilnessa, Zigale Hibstu Teffera, Habtamu Belew, Fassikaw Kebede.

**Supervision:** Tebelay Dilnessa, Mekuriaw Belayneh, Bewket Mesganaw, Adane Adugna, Desalegn Abebaw, Bantayehu Addis Tegegne, Fassikaw Kebede, Gashaw Azanaw Amare.

**Validation:** Abebe Fenta, Tebelay Dilnessa, Destaw Kebede, Adane Adugna, Wubetu Yihunie Belay, Fassikaw Kebede, Gashaw Azanaw Amare.

**Visualization:** Mekuriaw Belayneh, Bewket Mesganaw, Gashaw Azanaw Amare.

**Writing – original draft:** Abebe Fenta, Destaw Kebede.

**Writing – review & editing:** Abebe Fenta, Tebelay Dilnessa, Fassikaw Kebede, Gashaw Azanaw Amare.

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
