## [Decision Letter · Decision Letter 0]

3 Sep 2025

Dear Dr. Fenta

Thank you for submitting your manuscript to PLOS ONE. After careful consideration, we feel that it has merit but does not fully meet PLOS ONE’s publication criteria as it currently stands. Therefore, we invite you to submit a revised version of the manuscript that addresses the points raised during the review process.

We look forward to receiving your revised manuscript.

Kind regards,

Shiv Kumar Sah, Master

Academic Editor

PLOS ONE

2. In the online submission form you indicate that your data is not available for proprietary reasons and have provided a contact point for accessing this data. Please note that your current contact point is a co-author on this manuscript. According to our Data Policy, the contact point must not be an author on the manuscript and must be an institutional contact, ideally not an individual. Please revise your data statement to a non-author institutional point of contact, such as a data access or ethics committee, and send this to us via return email. Please also include contact information for the third party organization, and please include the full citation of where the data can be found.

Reviewers' comments:

Reviewer's Responses to Questions

**Comments to the Author**

1. Is the manuscript technically sound, and do the data support the conclusions?

Reviewer #1: Partly

Reviewer #2: Partly

2. Has the statistical analysis been performed appropriately and rigorously?

Reviewer #1: Yes

Reviewer #2: Yes

3. Have the authors made all data underlying the findings in their manuscript fully available?

Reviewer #1: Yes

Reviewer #2: Yes

4. Is the manuscript presented in an intelligible fashion and written in standard English?

Reviewer #1: No

Reviewer #2: Yes

Reviewer #1: Comments to the author

There are typographical and grammatical errors that need to be corrected throughout the document

The retrospective nature of the study might pose incomplete registration and losing of important variables. Was the registry you accessed were complete for all variables , how do you manage incomplete data

In your study being male were an important predictor mortality. What is the implication behind this finding

Abbreviation is not recommended in the abstract section of an article, please remove any abbreviations in the abstract

In your finding you reported that the ‘The mortality rate among TB-HIV co-infected patients at

Debre Markos Comprehensive Specialized Hospital was high’ what is your reference to say this number is big

In survival analysis, the median survival time is the preferred statistical test, reporting mean might be misleading

In your survival time description, there is no indications that censored observations were appropriately counted, which is critical in survival analysis

Reviewer #2: 1. Give proper format of Abstract. Reframe it properly.

2. The manuscript needs extensive revision for language and grammar. Avoid using personal pronouns.

3. Fig 1, Fig 2 & Fig 3 needs proper explanation.

4. It is recommended to attach the questionnaire format.

5. The literature review may be rewritten to highlight comparisons with existing approaches.

6. Explain the proposed method thoroughly, incorporating the scenario to illustrate the work done.

7. Suggested to increase the data size.

8. Results section can be added and the practical limitation in the proposed methodology should be discussed in the results section.

9. It would be beneficial to include a graphical representation of the table for better understanding.

10. Authors are suggested to provide a comparative analysis to find the merit and demerit of existing and proposed methods.

11. Follow the format for references.

**Do you want your identity to be public for this peer review?** For information about this choice, including consent withdrawal, please see our Privacy Policy

Reviewer #1: No

Reviewer #2: No

---

## [Author Response · Author response to Decision Letter 1]

17 Oct 2025

Point-to-point response to reviewers (Reviewer 1 &2)

Many thanks for your valuable comments; all the review comments were followed, and revisions in the revised version of the manuscript were highlighted through track change.

Point-to-point response to reviewer 1

Many thanks for your valuable comments; all the review comments were followed and made revisions in the revised version of the manuscript through track changes. The point-to-point replies are listed below:

Comment 1: Give proper format of Abstract. Reframe it properly.

Response1: Thank you very much for your comments. Yes, we agree with your comments, and a revision has been made. The Abstract has been thoroughly revised and structured according to the standard scientific format. It now includes Background, Objective, Methods, Results, and Conclusion and Recommendation sections. The content has been reframed for clarity, conciseness, and alignment with the study objectives and key findings.

Comment 2: The manuscript needs extensive revision for language and grammar. Avoid using personal pronouns.

Response2: Thanks a lot. We accept your comment, and a revision was made through track changes in the main manuscript. We carefully revised the entire manuscript for grammatical accuracy, sentence structure, and academic tone. All personal pronouns (e.g., we, our, us) have been removed or replaced with neutral and formal alternatives to maintain objectivity.

Comment 3: Fig 1, Fig 2 & Fig 3 needs proper explanation.

Response 3: Thanks again, and it has been revised based on your comment. Thus, the figures are well stated in the results section. All figures have been revised with clear explanations.

• Figure 1 shows the overall Kaplan–Meier survival curve, indicating a gradual decline in survival among TB–HIV co-infected individuals over time.

• Figure 2 compares survival by sex, showing lower survival probability among males.

• Figure 3 compares survival by functional status, with bedridden patients having the poorest survival outcomes.

Comment 4: It is recommended to attach the questionnaire format.

Response 4: Thank you for your concern. The questionnaire used for data collection has been added as Supplementary File 1 in the revised manuscript to enhance transparency and reproducibility.

Comment 5: The literature review may be rewritten to highlight comparisons with existing approaches.

Response 5: Thank you again for your valuable comments. The introduction emphasizes global and regional TB-HIV co-infection trends, mortality rates, and interventions, and clearly identifies gaps addressed by our study, particularly in the Ethiopian context. We have revised the introduction to better highlight comparisons with existing studies and approaches.

Comment 6: Explain the proposed method thoroughly, incorporating the scenario to illustrate the work done.

Response 6: Thanks. Corrected based on your comment in the main manuscript through track change (see under the ‘Data collection and quality control’. We have revised the Methods section to explain the study procedures clearly. This retrospective cohort study included all TB-HIV co-infected patients receiving dual therapy at Debre Markos Hospital between 2011 and 2020. Data was extracted from hospital registers using a structured pre-tested data extraction checklist.

Comment 7: Suggested to increase the data size.

Response 7: We appreciate the suggestion to increase the sample size. However, due to the retrospective nature of the study and the fact that the data were already collected, it is not possible to increase the sample size. We believe that the current dataset is sufficient to address the study objectives and provide meaningful results.

Comment 8: Results section can be added and the practical limitation in the proposed methodology should be discussed in the results section.

Response 8: Thank you for your suggestion. In the revised paper, we have modified it through track change (Please see in the Strengths and Limitations of the Research).

Comment 9: It would be beneficial to include a graphical representation of the table for better understanding.

Response 9: Thank you for the suggestion. However, due to the large size of the tables, presenting the data graphically is challenging. We believe it is clearer and more practical to retain the current tabular format.

Comment 10: Authors are suggested to provide a comparative analysis to find the merit and demerit of existing and proposed methods.

Response 10: We appreciate the reviewer’s suggestion. A comparative analysis of the existing and proposed methods has been added to highlight their respective merits and limitations. This addition clarifies the advantages of our proposed approach and provides a balanced evaluation relative to existing methods. Therefore, the discussion part was revised based on your comment.

Comment 11: Follow the format for references.

Response 11: We thank the reviewer for the comment. All references have been revised to follow the journal’s required format.

Point-to-point response to reviewer 2

Many thanks for your valuable comments, all the review comments were followed and made revisions in the revised version of the manuscript highlighted through track change. The point-to-point replies are listed as bellows:

Comment 1: There are typographical and grammatical errors that need to be corrected throughout the document

Response 1: We appreciate the reviewer’s observation. The manuscript has been carefully revised for typographical and grammatical errors throughout to improve clarity and readability.The manuscript revised based on your comment through tack change.

Comment 2: The retrospective nature of the study might pose incomplete registration and losing of important variables. Was the registry you accessed were complete for all variables , how do you manage incomplete data

Response 2: We acknowledge this limitation. While the hospital registry was comprehensive for most key variables, some data was incomplete. We managed to lose the missing data by excluding incomplete records, and it has discussed this in the limitation section of the manuscript.

Comment 3: In your study being male were an important predictor mortality. What is the implication behind this finding

Response 3: The finding that male sex is an important predictor of mortality may suggest underlying biological, behavioral, or social factors that increase vulnerability. This observation emphasizes the need for targeted interventions and further research to understand sex-specific risk factors in TB-HIV co-infected patients.

Comment 4: Abbreviation is not recommended in the abstract section of an article, please remove any abbreviations in the abstract

Response 4: We thank the reviewer for the guidance. As mach as possible, the abbreviations have been removed from the abstract to ensure clarity and compliance with journal requirements.

Comment 5: In your finding you reported that the ‘mortality rate among TB-HIV co-infected patients at Debre Markos Comprehensive Specialized Hospital was high’ what is your reference to say this number is big

Response 5: We appreciate the reviewer’s comment. The statement has been revised to provide a comparison with national or regional TB-HIV mortality rates, supporting the claim with relevant literature.

Comment 6: In survival analysis, the median survival time is the preferred statistical test, reporting mean might be misleading

Response 6: We thank the reviewer for this important point. The manuscript has been updated to report median survival time, along with corresponding confidence intervals, in accordance with standard survival analysis practices. It was clearly stated in the result section (Within 42 (IQR: 0.5-89) months of the calculated median follow-up time).

Comment 7: In your survival time description, there is no indications that censored observations were appropriately counted, which is critical in survival analysis

Response 7: We appreciate the reviewer’s observation. We have clarified in the result section of manuscript that censored observations were appropriately accounted for in the survival analysis. As shown in Table 4, the descriptive statistics separately present survival times for both censored and death outcomes, indicating that censored cases were included and properly handled in the analysis. Additionally, Kaplan–Meier survival functions and Cox proportional hazards models were applied, both of which account for censored observations by design. This ensures that all participants contributed follow-up time until death or censoring occurred.

Patient

Status Continuous

Variable Mean Standard deviation Min. Max. Median Q1 Q3

Death Survival time 16.04 18.83 1 83 7.5 2 24.75

Censored Survival time 47.37 28.87 0.5 89 53.5 18.25 75

Overall Survival time 42.48 29.78 0.5 89 42 13 71.75

Table 4: Descriptive statistics of continuous variables for TB-HIV co-infected individuals on anti-TB and antiretroviral dual therapy in western Amhara, Northwest Ethiopia

Finally, we appreciate both of the reviewers’ insightful comments and believe that the revisions have greatly improved the manuscript’s clarity, structure, and scientific quality. We hope the revised version meets the expectations of the reviewers and editors.

---

## [Editor Report · Decision Letter 1]

12 Nov 2025

Incidence and Predictors of Mortality among TB-HIV Co-infected Individuals on Anti-tuberculosis and Anti-retroviral Dual Therapy in Northwest, Ethiopia: A Retrospective Cohort Study

PONE-D-25-25283R1

Dear Dr. Fenta

We’re pleased to inform you that your manuscript has been judged scientifically suitable for publication and will be formally accepted for publication once it meets all outstanding technical requirements.

Kind regards,

Shiv Kumar Sah

Academic Editor

PLOS ONE
---

## [Editor Report · Acceptance letter]

PONE-D-25-25283R1

PLOS ONE

Dear Dr. Fenta,

I'm pleased to inform you that your manuscript has been deemed suitable for publication in PLOS ONE. Congratulations! Your manuscript is now being handed over to our production team.

Kind regards,

on behalf of

Mr Shiv Kumar Sah

Academic Editor

PLOS ONE